# Prognostic nutritional index as a prognostic factor for renal cell carcinoma: A systematic review and meta-analysis

Sung Ryul Shim[1], Sun Il Kim[2], Se Joong Kim[2], Dae Sung Cho[2]*

1 Department of Health and Medical Informatics, Kyungnam University College of Health Sciences, Changwon, Republic of Korea, 2 Department of Urology, Ajou University School of Medicine, Suwon, Korea

* urocho@aumc.ac.kr

## Abstract

### Background

Prognostic nutritional index (PNI) is a simple parameter which reflects patient's nutritional and inflammatory status and reported as a prognostic factor for renal cell carcinoma (RCC). Studies were included from database inception until February 2, 2022. The aim of this study is to evaluate prognostic value of PNI by meta-analysis of the diagnostic test accuracy in RCC.

### Methods and findings

Studies were retrieved from PubMed, Cochrane, and EMBASE databases and assessed sensitivity, specificity, summary receiver operating characteristic curve (SROC) and area under curve (AUC). Totally, we identified 11 studies with a total of 7,296 patients were included to evaluate the prognostic value of PNI in RCC finally. They indicated a pooled sensitivity of 0.733 (95% CI, 0.651–0.802), specificity of 0.615 (95% CI, 0.528–0.695), diagnostic odds ratio (DOR) of 4.382 (95% CI, 3.148–6.101) and AUC of 0.72 (95% CI, 0.68–0.76). Heterogeneity was significant and univariate meta-regression revealed that metastasis and cut-off value of PNI might be the potential source of heterogeneity. Multivariate meta-regression analysis also demonstrated that metastasis might be the source of heterogeneity.

### Conclusions

PNI demonstrated a good diagnostic accuracy as a prognostic factor for RCC and especially in case of metastatic RCC.

## Introduction

The ability to precisely predict the prognosis of patients with cancer is essential to determine the most appropriate treatment strategy and follow-up plan. Several prognostic factors for renal cell carcinoma (RCC) have been established or under estimation, including the

**Data Availability Statement:** All relevant data are within the manuscript and its Supporting information files.

**Funding:** The author(s) received no specific funding for this work.

**Competing interests:** The authors have declared
that no competing interests exist.

pathologic T stage, Fuhrman nuclear grade, tumor size, lymph node metastasis, and distant metastasis [1, 2]. Recently, there has been increasing evidence that nutritional status and the host immune response to RCC can significantly affect cancer progression and survival after treatment.

The prognostic nutritional index (PNI), a novel method to assess immune and nutritional status on the basis of the serum lymphocyte count and albumin level, has been introduced as a simple tool with prognostic value for patients with RCC [3–13]. However, insufficient results have been reported regarding the utility of the PNI in patients with RCC, due to differences among studies in sample size, presence of metastasis, patient characteristics, and other factors [14]. Therefore, we performed this pooled meta-analysis to assess the diagnostic accuracy of the PNI as a prognostic factor for RCC based on available outcome data.

The aim of current study is to complete meta-analysis for diagnostic accuracy of the PNI for the prediction of survival in RCC patients, in order to provide more evidence-based data of PNI as a prognostic factor in RCC patients.

## Methods

This systematic review and meta-analysis was performed in accordance with the standard Preferred Reporting Items for Systematic Reviews and Meta-Analyses of Diagnostic Test Accuracy Studies (PRISMA-DTA) guidelines and was registered to the International Prospective Register of Systematic Reviews (registration no. CRD42020185171) [15].

### Literature search

Based on a standardized protocol, a systemic, comprehensive search of the PubMed, Web of Science, the Cochrane Library, and EMBASE databases was conducted to identify studies that evaluated the prognostic value of the PNI in patients with RCC. Studies were included from database inception until February 2, 2022. Searches were performed using the following MeSH terms and keywords: "RCC", "renal cancer," "carcinoma," "renal cell," "kidney cancer," "kidney neoplasms," "clear cell carcinoma," "adenocarcinoma, clear cell," "non-clear cell carcinoma", "prognostic nutritional index", "PNI", "prognosis", "survival" and "outcome". Reference lists of retrieved articles were checked to identify additional studies. Abstracts and conference proceedings were also included in the literature search.

### Study selection and definition

Initial screening of search results based on titles and abstracts was performed based on structured questions using the PICO methodology; Populations: patients with RCC; Intervention: high PNI value; Comparator: low PNI value; Outcomes: survival; Decisions regarding study eligibility for inclusion in the meta-analysis, based on full-text review, were performed independently by two reviewers (SIK and DSC). Disagreements regarding data extraction and methodological assessment were resolved by discussion; remaining disagreements were resolved by a third reviewer (SJK). Each included study was carefully checked to ensure that no duplicate data were included in the meta-analysis. Studies were considered eligible for inclusion if they met the following criteria: (1) PNI values were obtained before treatment, and numbers of patients were reported according to PNI cutoff values; (2) treatments were limited to surgery, targeted therapy, or immunotherapy; and (3) the relationship between RCC prognosis and PNI value was analyzed. Papers written in languages other than English were included if the data could be extracted. Letters, review articles, and case reports were excluded. When data from the same patients were reported in more than one article, only the most recent article was included in the analysis.

The PNI was defined based on the serum albumin level and lymphocyte count using the following formula: $10 \times$ serum albumin (g/dL) $+ 0.005 \times$ total lymphocyte count of peripheral blood (per mm$^3$) [16].

### Data extraction

The following information was extracted from the studies: (1) study attributes, including author names, year of publication, region, research period, and sample size; (2) patient characteristics, including age, sex, and follow-up duration; (3) RCC characteristics, including tumor type, stage, and distant metastasis; (4) PNI values; and (5) survival outcomes, including cancer-specific survival and/or overall survival. The absolute numbers of true-positive, true-negative, false-positive, and false-negative cases were extracted or calculated, and then incorporated into a $2 \times 2$ contingency table.

### Statistical analysis

Pooled estimates of sensitivity and specificity and their 95% confidence intervals (CI) were calculated as the main outcome measures and were analyzed by forest plots. We used the bivariate random-effects model for analysis and pooling of the diagnostic performance measures across studies. The threshold effect was assessed using the receiver operating characteristic (ROC) plane and the Spearman correlation coefficient. The ROC plane is the graphic representation of the pairs of sensitivity and specificity, and it characteristically shows a curvilinear pattern if a threshold effect exists. Study heterogeneity was measured using the Cochran's Q and $I^2$ tests; $p < 0.10$ and $I^2 > 50\%$ were considered to indicate significant heterogeneity. Study heterogeneity was calculated using subgroup and meta-regression analyses were conducted to identify potential sources of heterogeneity. Publication bias was determined based on the degree of asymmetry in Deeks funnel plots. All statistical analyses were performed using Stata (version 14.0; StataCorp, College Station, TX, USA), and Meta-DiSc software (version 1.4; Meta-DiSc, Madrid, Spain). A p-value $< 0.05$ was considered to indicate statistical significance.

### Risk of bias and quality assessment

Two investigators (SIK and DSC) independently assessed all included studies for methodological quality and potential sources of bias using the Quality Assessment of Diagnostic Accuracy Studies-2 (QUADAS-2) tool in Stata software [17]. Any disagreements regarding the appropriate category for a study were resolved by discussion.

## Results

### Search results and study characteristics

Fifty-three studies were identified in the initial database search. Following review of the titles and abstracts, 19 articles were identified that analyzed the relationship between RCC and the PNI. From among these 19 articles, 11 retrospective studies of 7,296 RCC patients were included in the meta-analysis [3–13]. The search strategy is presented in Fig 1. The main reasons for study exclusion were a lack of focus on the PNI when diagnosing RCC, or an absence of information regarding the PNI.

Table 1 shows the characteristics of the eleven included studies, all of which were retrospective in nature. The studies originated from various countries, including Austria, the United States, China, Turkey, and Korea. Six studies enrolled $\geq 350$ patients and five had $< 350$ patients. Seven studies included patients with non-metastatic RCC; the remaining four included metastatic RCC patients. Cut-off values for the PNI differed among studies.

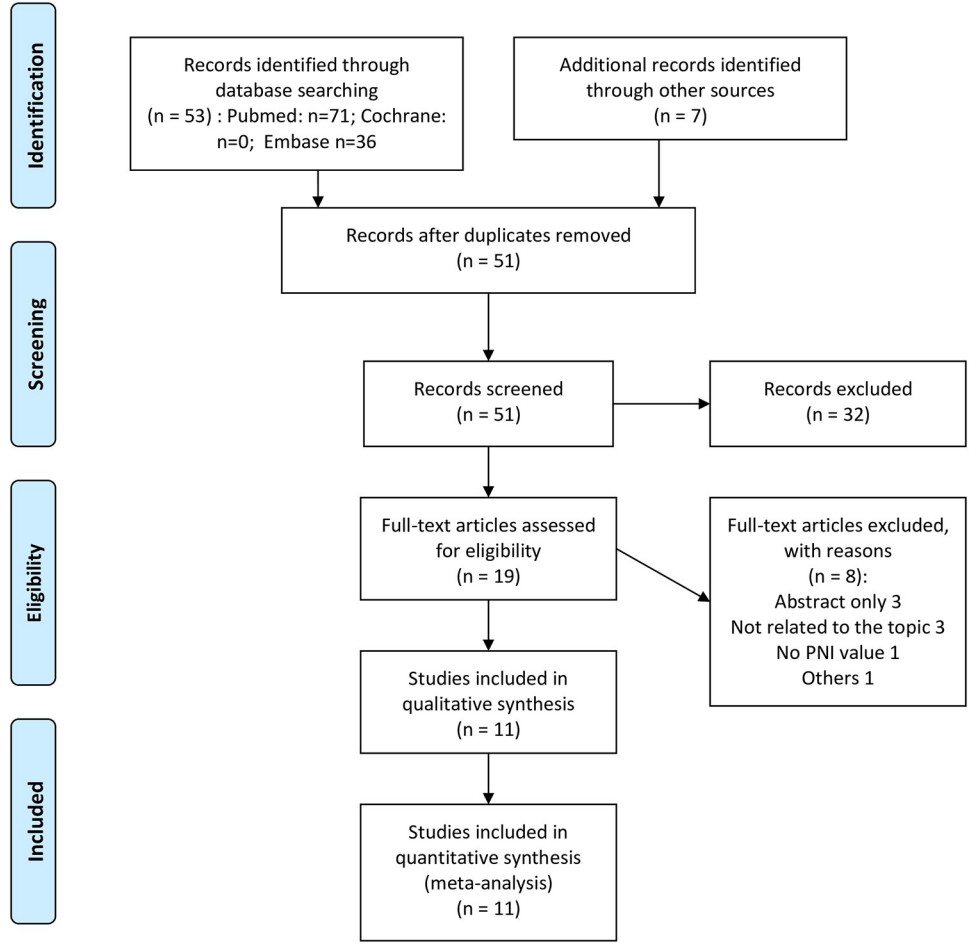

**Fig 1. Flow diagram of studies identified in meta-analysis.** PNI indicates Prognostic Nutritional Index.

## Methodological quality assessment and risk of bias

Methodological quality assessment and risk of bias were evaluated using QUADAS-2; the results are summarized in Fig 2. Visual inspection of a Deeks' funnel plot indicated asymmetry, i.e., significant publication bias or small-study effects (p = 0.02, S1 Fig).

## Quantitative synthesis

The data of the eleven studies that examined the prognostic value of the PNI for RCC were pooled. The pooled sensitivity was 0.733 (95% CI, 0.651–0.802) and the pooled specificity was 0.615 (95% CI, 0.528–0.695) (Fig 3). The diagnostic odds ratio was 4.382 (95% CI, 3.148–6.101) and the area under the curve was 0.72 (95% CI, 0.68–0.76) (Fig 4). The ROC curve to analyze the relationship between sensitivity and specificity was symmetrical, whereby the diagnostic odds ratio did not vary along the curve (p = 0.615). Furthermore, the ROC plot confirmed the absence of a threshold effect. Substantial heterogeneity among all of the included studies was observed in terms of sensitivity (Cochran's Q = 253.64, p<0.001, $I^2$ = 96.06%) and specificity (Cochran's Q = 97.32, p<0.001, $I^2$ = 89.73%).

**Table 1. Characteristics of all included studies.**

| Study cohort | Year | Study region | Research time | Follow-up (month) | M/F (n) | Age (years) | Tumor type | Distant metastasis (n) | PNI value | TP | FP | FN | TN | Sensitivity (95% CI) | Specificity (95% CI) |
|---|---|---|---|---|---|---|---|---|---|---|---|---|---|---|---|
| Hofbauer et al | 2015 | Austria and USA | 1991–2012 | Median: 40 | 892/452 (1344) | Median (IQR): 62 (53–70) | RCC | 399 | Median (IQR): 50.6 (45.8–54.6); Cut-off: 48 | 423 | 440 | 142 | 339 | 0.749 (0.711–0.784) | 0.435 (0.400–0.471) |
| Broggi et al | 2016 | USA | 2001–2014 | NA | 204/115 (319) | Median: 61.5 | Clear cell RCC | 0 | Mean (SD): 44.2 (6.7); Cut-off: 44.7 | 109 | 33 | 80 | 71 | 0.577 (0.503–0.648) | 0.683 (0.584–0.771) |
| Jeon et al | 2016 | South Korea | 1994–2008 | Mean (range): 68.6 (1.2–212.6) | 1011/426 (1437) | Mean (range): 54.2 (20–85) | RCC | 106 | Mean (range): 52.7 (27.7–85.3); Cut-off: 51 | 922 | 38 | 396 | 81 | 0.700 (0.674–0.724) | 0.681 (0.589–0.763) |
| Kwon et al | 2017 | South Korea | 2007–2014 | Median (IQR): 45.3 (23.7–77.3) | 99/26 (125) | Median (IQR): 58 (51–66) | Metastatic RCC | 125 | Median (IQR): 42.0 (37.2–45.1); Cut-off: 41 | 24 | 44 | 5 | 52 | 0.828 (0.642–0.942) | 0.542 (0.437–0.644) |
| Kang et al | 2017 | South Korea | 1996–2012 | Mean: 79.6 | 241/83 (324) | Median (IQR): 55 (48–64) | RCC | 0 | Median (IQR): 45.0 (42.01–46.51); Cut-off: 45 | 157 | 6 | 134 | 27 | 0.540 (0.480–0.598) | 0.818 (0.645–0.930) |
| Peng et al | 2017 | China | 2001–2010 | Median (IQR): 67 (2–108) | 952/408 (1360) | Median (IQR): 55 (14–87) | RCC | 61 | NA, Cut-off: 47.625 | 939 | 39 | 317 | 65 | 0.748 (0.723–0.771) | 0.625 (0.525–0.718) |
| Cai et al | 2017 | China | 2006–2015 | Median (IQR): 22 | 135/43 (178) | Median (IQR): 60 (24–82) | Metastatic RCC | 178 | Median (IQR): 52.3 (21.6–88.8); Cut-off: 51.62 | 61 | 37 | 10 | 70 | 0.859 (0.756–0.930) | 0.654 (0.556–0.744) |
| Yasar et al | 2019 | Turkey | 2007–2017 | NA | 258/138 (396) | Median (IQR): 58 (29–88) | Metastatic RCC | 396 | Median (IQR): 38.5 (18–52); Cut-off: 38.5 | 81 | 75 | 33 | 124 | 0.711 (0.618–0.792) | 0.623 (0.552–0.691) |
| Cho et al | 2020 | South Korea | 1994–2017 | Median (IQR): 72 (4–272) | 307/152 (459) | Mean (range): 55.8 (18–81) | RCC | 0 | Median (IQR): 53.0 (30.9–69.0); Cut-off: 51 | 295 | 0 | 154 | 10 | 0.657 (0.611–0.701) | 1.000 (0.692–1.000) |
| Hu et al | 2020 | China | 2010–2013 | Median (IQR): 83 (74–93) | 256/404 (660) | Mean: 54.89 | RCC | 18 | Median (IQR): 51.05 (47.9–53.88); Cut-off: 44.3 | 550 | 41 | 46 | 23 | 0.921 (0.901–0.942) | 0.362 (0.241–0.490) |
| Tang et al | 2021 | China | 2009–2014 | Median (IQR): 60.9 (46.9–76.1) | 442/252 (694) | NA | RCC | 0 | Cut-off: 49.075 | 406 | 21 | 244 | 23 | 0.622 (0.591–0.659) | 0.521 (0.370–0.681) |

TP, true positive; FP, false positive; FN, false negative; TN, true negative; CI, confidence interval; NA, not applicable; RCC, renal cell carcinoma

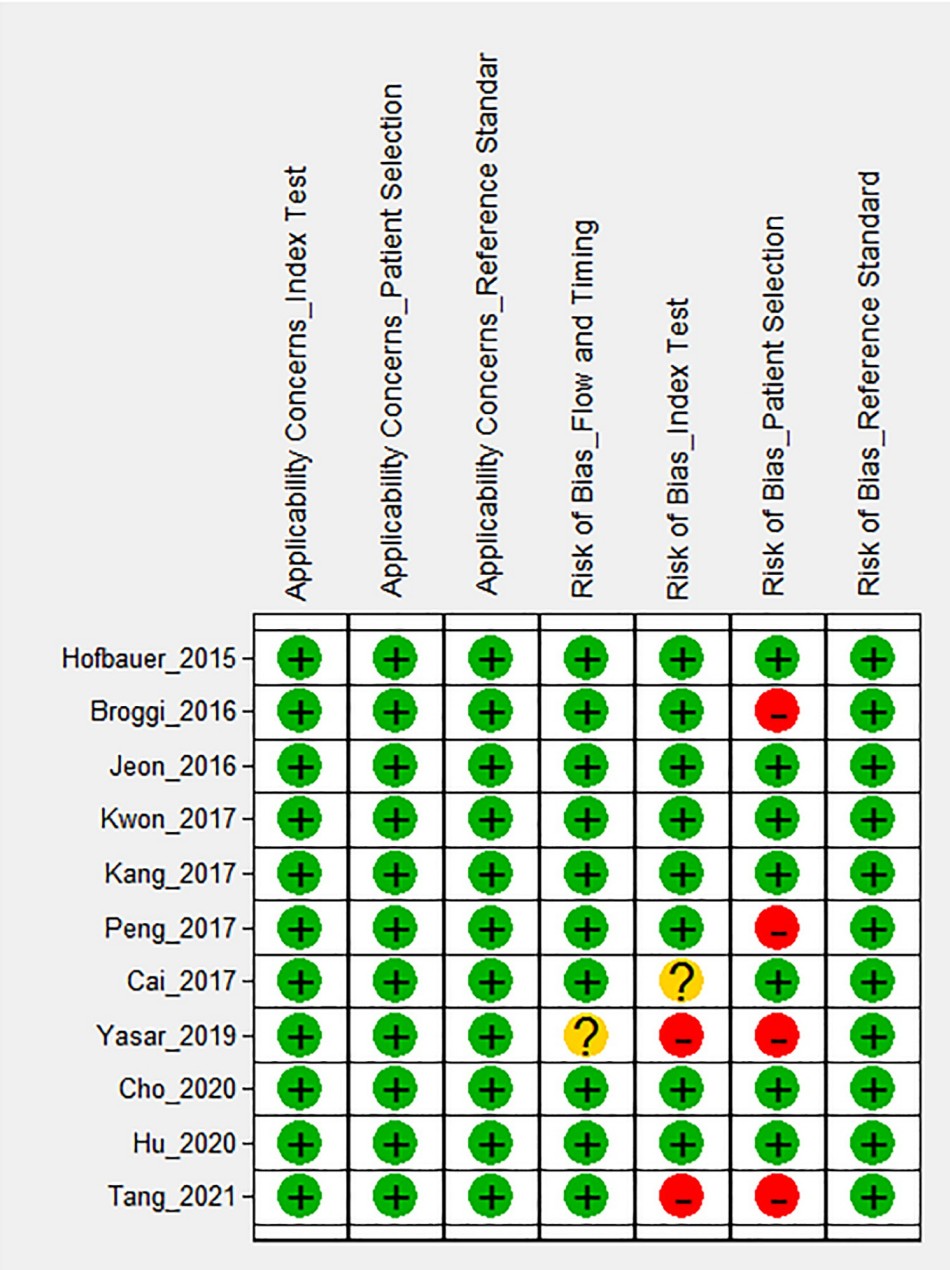

**Fig 2. Summary of the methodological quality of the studies evaluated by the quality assessment of diagnostic accuracy studies-2 (QUADAS-2).**

## Subgroup and meta-regression analyses

Subgroup analyses were performed to identify potential sources of heterogeneity in the diagnostic accuracy of the PNI among studies, including ethnicity (Asian vs. Caucasian), sample size (n ≥ 350 vs. n < 350), presence of metastasis, PNI cut-off value (≥ 50 vs. < 50), QUADAS-2 classification (low risk vs. high risk), and proportion of males (≥ 70% vs. < 70%) (Table 2). In addition, we performed subgroup analysis with American Society of

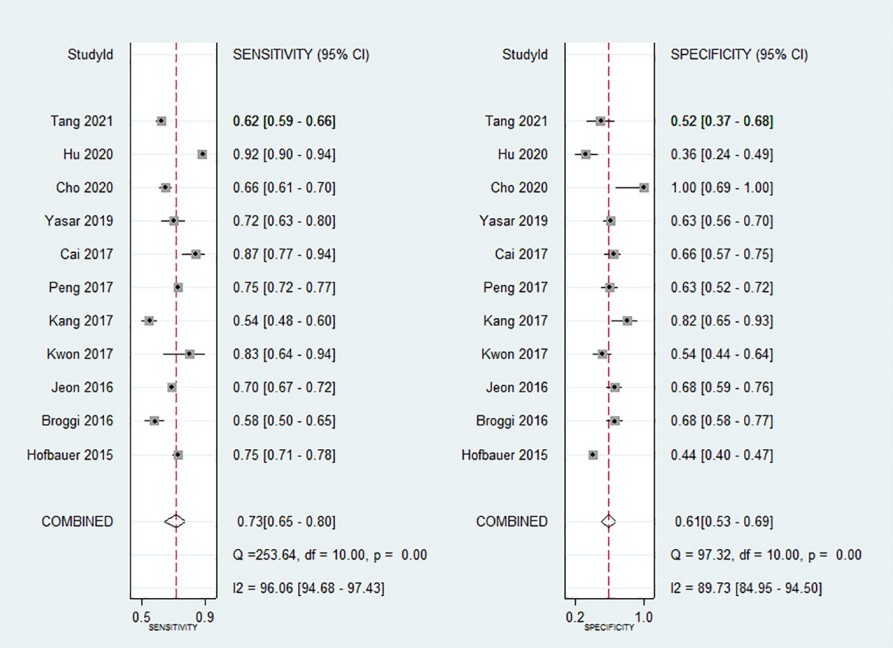

**Fig 3. Forrest plot of the sensitivity and specificity of the prognostic nutritional index as prognostic value for renal cell carcinoma.** CI indicates confidence interval.

Anesthesiologists (ASA) or Eastern Cooperative Oncology Group (ECOG) status. As a result of statistical analysis, there was no significant difference between ASA >2 group and ASA ≤2 group. In addtion, there were only 4 out of 11 studies showing ASA or ECOG score and limitation to the analysis. Subgroup analyses showed that presence of metastasis and PNI cut-off value affected the diagnostic accuracy of the PNI for RCC. In univariate meta-regression analysis, the sensitivity and specificity were 0.85 and 0.55, respectively, in the metastatic group, and 0.66 and 0.65, respectively, in the non-metastatic group (p = 0.01). Also, the sensitivity and specificity were 0.74 and 0.72, respectively, in the PNI ≥ 50 group, and 0.73 and 0.57, respectively, in the PNI < 50 group (p = 0.05). Multivariate meta-regression analysis demonstrated significant differences in sensitivity and specificity between the metastatic RCC and non-metastatic RCC groups (p = 0.035).

## Discussion

To the best of our knowledge, this meta-analysis is the first to assess the diagnostic accuracy of the PNI as a prognostic factor for RCC. This study suggests that the PNI has value as a prognostic factor for RCC. Therefore, the PNI can aid clinicians in predicting the clinical outcomes of RCC and patients with low PNI need to be managed by nutritional support and treated in a way to correct malnutritional status. Because there was heterogeneity among the studies included in our meta-analysis, subgroup and univariate meta-regression analyses were also performed based on ethnicity, sample size, presence of metastasis, PNI cut-off values, QUADAS-2 classification, and sex ratio. These analyses showed that presence of metastasis and PNI cut-off values were potential sources of heterogeneity among the included studies. In addition, presence of metastasis (p = 0.035) were significant sources of heterogeneity in the diagnostic performance of the PNI in multivariate meta-regression analysis. These findings suggested

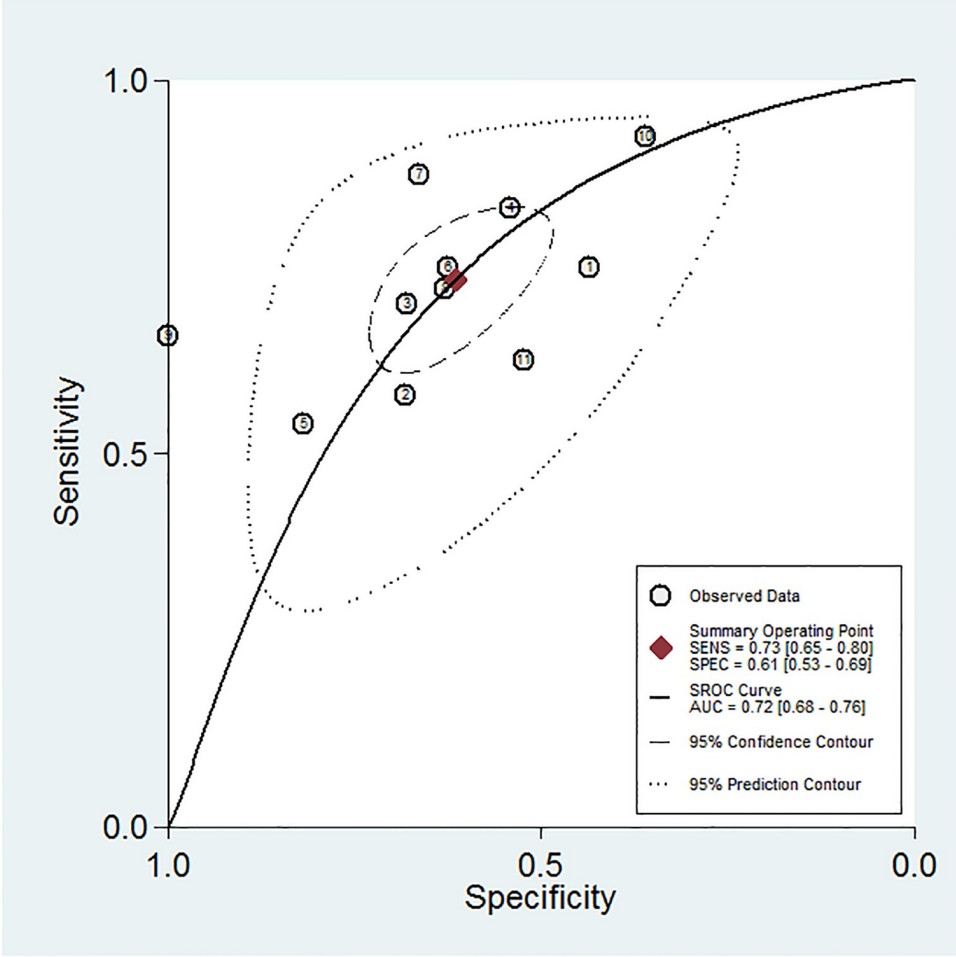

**Fig 4. Summary receiver operating characteristic graph for the included studies.** AUC = area under curve; SENS = sensitivity; SPEC = specificity; SROC = summary receiver operating characteristic.

that the PNI clearly had superior prognostic value in patients with metastatic RCC compared with non-metastatic RCC. Thus, it will be important to determine the utility of new prognostic scoring systems for patients with RCC based on the PNI, especially in case of metastatic RCC.

Several prognostic factors and models have been proposed to predict the clinical outcomes of RCC, including the tumor-node-metastasis (TNM) staging system and the Fuhrman nuclear grade. However, patients with the same TNM stage or Fuhrman grade may have significantly different prognostic courses [18]. Therefore, many studies have attempted to identify additional factors that can precisely predict the prognosis of RCC. The PNI was first used by Onodera et al. [16] to evaluate the inflammation and nutritional status of patients who underwent gastrointestinal surgery; this simple index is calculated from the serum albumin level and lymphocyte count. Because laboratory assessments, including the serum albumin level and lymphocyte count, are routinely performed before treatment of patients with RCC, PNI values can be easily measured. As reported previously [11], we observed a strong inverse relationship between the PNI and tumor aggressiveness, and a lower PNI was also associated with poorer patient outcomes. This suggests the potential for a significant association among the PNI, pathological characteristics of RCC, and other known risk factors for RCC. Additional studies

**Table 2. Univariate and multivariate meta-regression analysis for identifying potential sources of heterogeneity in the diagnostic performance of screening tests.**

| Variable | No. of studies | Univariate* | | | Multivariate† | |
|---|---|---|---|---|---|---|
| | | Sensitivity | Specificity | p-value | Diagnostic OR (95% CI) | p-value |
| Ethnicity | | | | | | |
| Asian | 8 | 0.75 | 0.64 | 0.19 | 1.14 (0.56–2.30) | 0.636 |
| Caucasian | 3 | 0.68 | 0.58 | | | |
| No. of patients | | | | | | |
| ≥350 | 7 | 0.75 | 0.57 | 0.40 | 0.85 (0.44–1.64) | 0.522 |
| <350 | 4 | 0.71 | 0.68 | | | |
| Tumor type | | | | | | |
| Metastasis | 4 | 0.85 | 0.55 | 0.01 | 1.99 (1.08–3.65) | 0.035 |
| Non-metastasis | 7 | 0.66 | 0.65 | | | |
| PNI cut-off value | | | | | | |
| ≥50 | 3 | 0.74 | 0.72 | 0.05 | 1.24 (0.61–2.54) | 0.448 |
| <50 | 8 | 0.73 | 0.57 | | | |
| QUADAS-2 | | | | | | |
| Low risk | 6 | 0.68 | 0.62 | 0.25 | 0.94 (0.38–1.47) | 0.258 |
| High risk | 5 | 0.79 | 0.60 | | | |
| Men, % | | | | | | |
| ≥70 | 5 | 0.74 | 0.67 | 0.12 | 1.65 (0.71–3.86) | 0.177 |
| <70 | 6 | 0.73 | 0.55 | | | |

PNI: Prognostic Nutritional Index, OR: odds ratio, CI: confidence interval,

*: analyzed by STATA,

†: analyzed by Meta-Disc, Univariate p-value of joint model for sensitivity and specificity

are needed to more clearly elucidate how the PNI is related to the prognosis of patients with RCC.

Recently, prognostic role of circulating biomarkers associated with different features of RCC biology has been proposed, including carbonic anhydrase IX (CAIX), hypoxia-inducible factor-1α (HIF1α), CA15-3, PTX3, and C-reactive protein (CRP) [19–22]. These biomarkers are suggested to be related to the prognosis of RCC. In addition, RCC is a metabolic disease characterized by a reprogramming of energetic metabolism. In particular the metabolic flux through glycolysis is partitioned and mitochondrial bioenergetics and OxPhox are impaired, as well as lipid metabolism [23–27]. A recent study also delineated a lipidomic profile of human clear cell RCC and integrated it with transcriptomic data to connect the variations in cancer lipid metabolism with gene expression changes [28].

This meta-analysis had several limitations. First, it included relatively few studies (N = 11). Therefore, validation of the results is needed via meta-analyses including more studies. Furthermore, this meta-analysis obviously could not consider unpublished data. Second, there was considerable heterogeneity in the pooled estimates; despite attempts to determine the sources of heterogeneity through meta-regression, a substantial proportion of the variance remained unexplained, and many factors could not be assessed because they were not reported in all of the studies. Furthermore, the small number of included studies limited the statistical power of the multivariate meta-regression. Third, individual patient characteristics (e.g., comorbidities, alcohol consumption, smoking history, and obesity) were not considered, although these may affect the PNI by inducing systemic inflammation or altering nutritional status. Fourth, randomized controlled trial and high-level studies were not included in this

study and that undermined the value of this study. In conclusion, the results of this study demonstrate that diagnostic accuracy of the PNI as a prognostic factor for patients with RCC, especially metastatic RCC. In addition, the PNI is a simple, cost-effective, and widely available tool. Therefore, new prognostic scoring systems that include the PNI could be useful for predicting the prognosis of patients with RCC.

## Supporting information

**S1 Fig. Deeks' funnel plot for asymmetry test for detecting publication bias.**
(TIF)

**S1 File.**
(XLSX)

**S2 File.**
(XLSX)

**S1 Checklist.**
(DOC)

## Author Contributions

**Conceptualization:** Sung Ryul Shim, Sun Il Kim, Dae Sung Cho.

**Data curation:** Sung Ryul Shim, Sun Il Kim, Se Joong Kim, Dae Sung Cho.

**Formal analysis:** Sung Ryul Shim, Dae Sung Cho.

**Methodology:** Dae Sung Cho.

**Writing – original draft:** Sun Il Kim, Dae Sung Cho.

**Writing – review & editing:** Se Joong Kim, Dae Sung Cho.

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
