## [Decision Letter · Decision Letter 0]

29 May 2022

PONE-D-22-03642Diagnostic Accuracy of Prognostic Nutritional Index as a Prognostic Factor for Renal Cell Carcinoma: A Systematic Review and Meta-AnalysisPLOS ONE

Dear Dr. Cho,

Thank you for submitting your manuscript to PLOS ONE. After careful consideration, we feel that it has merit but does not fully meet PLOS ONE’s publication criteria as it currently stands. Therefore, we invite you to submit a revised version of the manuscript that addresses the points raised during the review process.

We look forward to receiving your revised manuscript.

Kind regards,

Giuseppe Lucarelli, M.D., Ph.D.

Academic Editor

PLOS ONE

Journal Requirements:

Reviewers' comments:

Reviewer's Responses to Questions

**Comments to the Author**

1. Is the manuscript technically sound, and do the data support the conclusions?

Reviewer #1: Yes

Reviewer #2: Partly

Reviewer #3: Yes

2. Has the statistical analysis been performed appropriately and rigorously? 

Reviewer #1: Yes

Reviewer #2: Yes

Reviewer #3: Yes

3. Have the authors made all data underlying the findings in their manuscript fully available?

Reviewer #1: Yes

Reviewer #2: Yes

Reviewer #3: Yes

4. Is the manuscript presented in an intelligible fashion and written in standard English?

Reviewer #1: Yes

Reviewer #2: No

Reviewer #3: Yes

5. Review Comments to the Author

Reviewer #1: Objectives: Authors aim for this systematic review and meta-analysis is to evaluate prognostic value of Prognostic Nutritional index by meta-analysis of the diagnostic test accuracy in kidney cancer.

There are some minor issues that need to be corrected before the manuscript can be published.

1. The title of the introduction section should be modified (Itroduction)

2. The calculation formula of PNI is incorrect. I would ask the authors to correct it.

3. Figure 2 legend has a mistype error (“qualityof”)

4. In table 1 the authors wrote distal metastasis. Probably the authors meant distant. If it is the case, I would like to ask the authors to correct it.

5. In Figure 4 legend I believe that the acronym authors had intended to use is “SPEC” not “SEPC”. I would ask the authors to correct it.

Reviewer #2: In the current manuscript the authors evaluated the diagnostic accuracy of the prognostic nutritional index (PNI) for renal cell carcinoma (RCC) in a pooled-analysis fashion. Overall the topic is interesting and timely, however, it is not new. Other groups reported SR & MA on PNI (DOI: 10.3389/fonc.2021.719941 - DOI: 10.1016/j.urolonc.2021.05.028) reaching the conclusion that PNI could be considered a potential prognostic predictor of treatment outcomes for patients with RCC. Neverthless, the authors of the current SR tried to slice the topic evaluating the diagnostic accuracy level only.

Please find below some observation from my side.

Abstract. Please remove the sentences "However, the prognostic value of PNI in RCC remains unclear.". It is absolutely not supported by the available literature.

Abstract. Could authors clarify the period of paper collecting.

Introduction. This sentence " Several prognostic factors for renal cell carcinoma (RCC) have been established, including the pathologic T stage, Fuhrman nuclear grade, tumor size, lymph node metastasis, and distant metastasis" needs strong references, indeed, the role of some of cited parameters are still under debate.

Introduction. Could the author add a reference here "However, conflicting results have been reported regarding the utility of the PNI in patients with RCC, due to differences among studies in sample size, presence of metastasis, patient characteristics, and other factors."

Introduction. A clear sentence about the aims of this study lacks.

M&M. Please the authors provide the PICOS.

Could the authors explain why they decided to include papers written in languages other than English. How were the articles traslated?

M&M. The statistical analysis is reported in a remarkable way! Howeveer, the lack of RCT and High-level studies is a crucial negative-point.

Results. Did the authors consider to evaluate ASA >2 patients as subgrup for the analysis? Please the authors report how to decide the variable for subgroup analysis.

Discussion. Please absolutEly remove this sentences since they are not point of streight, but the normal way for addressing a SR-MA "The strengths of this study included the comprehensive literature search strategy based on a standardized protocol. Furthermore, rigorous data analysis methods were applied, such as bivariate random-effects meta-analysis (including covariates) and ROC curve analysis."

Discussion. I believe that the discussion is the place where discuss about the results of analysis in a critical way. It is not the place where cut and paste the results of other work or talk about random arguments. Please handle it again.

Conclusion. The conclusions are not supported by the results. They are not in line with the study aim that are to demonstrate the diagnostic accuracy of PNI as a prognostic factor for RCC

References. They are up-to-date. However, several sentences in the manuscript lack references.

Table and Figures. They are of good quality.

Please a native speacker check is straight reccomanded.

Reviewer #3: In this meta-analysis, the authors evaluated the role of the PNI as a prognostic factor for RCC.

I have some comments:

- Please rephrase the title and the text when you use the term "diagnostic" in association with "prognostic". For example in the title "diagnostic accuracy...as prognostic factor". Diagnosis and prognosis are different processes of medical evaluation. Please remove the term "diagnostic", in this study it has been evaluated the prognostic role of PNI.

-A prognostic role has been proposed for other circulating biomarkers associated with different features of RCC biology, including carbonic anhydrase IX (CAIX), hypoxia-inducible factor-1α (HIF1α), CA15-3, PTX3, and C-reactive protein (CRP) (ref:PMID: 15126876; PMID: 24692843; PMID: 32345771;PMID: 20006861)

These studies should be referenced and discussed.

-RCC is a metabolic disease characterized by a reprogramming of energetic metabolism. In particular the metabolic flux through glycolysis is partitioned (PMID: 30983433, PMID: 29371925, PMID: 28933387; PMID: 30538212), and mitochondrial bioenergetics and OxPhox are impaired , as well as lipid metabolism (PMID: 30538212; PMID: 32861643). In addition a recent study (PMID: 33322148) delineated a lipidomic profile of human ccRCC and integrated it with transcriptomic data to connect the variations in cancer lipid metabolism with gene expression changes. These findings should be referenced and discussed.

6. PLOS authors have the option to publish the peer review history of their article (what does this mean?). If published, this will include your full peer review and any attached files.

Reviewer #1: **Yes: **Tataru Octavian Sabin

Reviewer #2: No

Reviewer #3: No

---

## [Author Response · Author response to Decision Letter 0]

6 Jul 2022

Thank you for your constructive comments and suggestions. We reviewed our manuscript and did all our best to revise the manuscript as you suggested. Those are as follows:

Reviewer #1:

1) The title of the introduction section should be modified (Itroduction)

Answer

We thank the reviewer for this comment. We changed “Itroduction” to “Introduction” as reviewer suggested.

2) The calculation formula of PNI is incorrect. I would ask the authors to correct it.

Answer

We thank the reviewer for this important comment. We corrected the calculation formula of PNI as reviewer suggested.

3) Figure 2 legend has a mistype error (“qualityof”)

Answer

We thank the reviewer for this comment. We changed “qualityof” to “quality of” as reviewer indicated.

4) In table 1 the authors wrote distal metastasis. Probably the authors meant distant. If it is the case, I would like to ask the authors to correct it.

Answer

We thank the reviewer for this comment. We changed “distal” to “distant” as reviewer suggested. 

5) In Figure 4 legend I believe that the acronym authors had intended to use is “SPEC” not “SEPC”. I would ask the authors to correct it..

Answer

We thank the reviewer for this comment. We changed “SEPC” to “SPEC” as reviewer indicated.

Reviewer #2:

1) Abstract. Please remove the sentences "However, the prognostic value of PNI in RCC remains unclear.". It is absolutely not supported by the available literature.

Answer

We thank the reviewer for this comment. We removed the sentences "However, the prognostic value of PNI in RCC remains unclear." In the abstract part of manuscript.

2) Abstract. Could authors clarify the period of paper collecting?

Answer

We thank the reviewer for this comment. To comply with the reviewer’s recommendations, we clarified the period of paper collecting in the abstract part of manuscript.

3) Introduction. This sentence " Several prognostic factors for renal cell carcinoma (RCC) have been established, including the pathologic T stage, Fuhrman nuclear grade, tumor size, lymph node metastasis, and distant metastasis" needs strong references, indeed, the role of some of cited parameters are still under debate.

Answer

We thank the reviewer for this important comment. To comply with the reviewer’s recommendation, we added references. In addition, we revised the sentence “have been established” to “have been established or under estimation”.

4) Introduction. Could the author add a reference here "However, conflicting results have been reported regarding the utility of the PNI in patients with RCC, due to differences among studies in sample size, presence of metastasis, patient characteristics, and other factors."

Answer

We thank the reviewer for this pertinent comment. To comply with the reviewer’s recommendations, we added references. In addition, we revised the term “conflict” to “insufficient”.

5) Introduction. A clear sentence about the aims of this study lacks.

Answer

We thank the reviewer for this important comment. To comply with the reviewer’s recommendations, we inserted the sentence “The aim of current study is to complete meta-analysis for diagnostic performance of PNI for the prediction of recurrence or survival in RCC patients, in order to provide more evidence-based data of PNI as a prognostic factor in RCC patients.” In the introduction part of manuscript.

6) M&M. Please the authors provide the PICOS.

Answer

We thank the reviewer for this pertinent comment. To comply with the reviewer’s recommendations, we inserted the sentence “Initial screening of search results based on titles and abstracts was performed based on structured questions using the PICO methodology: Populations: patients with RCC; Intervention: high PNI value; Comparator: low PNI value; Outcomes: survival” In the Methods part of manuscript.

7) Could the authors explain why they decided to include papers written in languages other than English. How were the articles traslated?

Answer

We thank the reviewer for this important comment. We tried to include all possible data because there were only a few papers about this topic. Therefore, we tried to include papers written in lanaguages other than English but unfortunately there was no suitable paper included in this study.

8) M&M. The statistical analysis is reported in a remarkable way! Howeveer, the lack of RCT and High-level studies is a crucial negative-point.

Answer

We thank the reviewer for this pertinent comment. We totally agree with the reviewer’s opinion. Therefore, we described it as a limitation point of this study in the discussion part of manuscript.

9) Results. Did the authors consider to evaluate ASA >2 patients as subgrup for the analysis? Please the authors report how to decide the variable for subgroup analysis..

Answer

We thank the reviewer for this important comment. To comply with the reviewer’s recommendations, we performed subgroup analysis with ASA >2 patients. As a result of statistical analysis, there was no significant difference between ASA >2 group and ASA ≤2 group. In addtion, there were only 4 out of 11 studies showing ASA or ECOG score and limitation to the analysis. We described it in the result part of manuscript. We tried to perform subgroup analysis with variables for which data from all papers were available. As a result, these variables such as ethnicity, sample size, presence of metastasis, PNI cut-off value, QUADAS-2 classification, and proportion of males were included.

10) Discussion. Please absolutEly remove this sentences since they are not point of streight, but the normal way for addressing a SR-MA "The strengths of this study included the comprehensive literature search strategy based on a standardized protocol. Furthermore, rigorous data analysis methods were applied, such as bivariate random-effects meta-analysis (including covariates) and ROC curve analysis."

Answer

We thank the reviewer for this pertinent comment. To comply with the reviewer’s recommendations, we removed the sentences "The strengths of this study included the comprehensive literature search strategy based on a standardized protocol. Furthermore, rigorous data analysis methods were applied, such as bivariate random-effects meta-analysis (including covariates) and ROC curve analysis." In the discussion part of the manuscript.

11) Discussion. I believe that the discussion is the place where discuss about the results of analysis in a critical way. It is not the place where cut and paste the results of other work or talk about random arguments. Please handle it again.

Answer

We thank the reviewer for this important comment. To comply with the reviewer’s recommendation, we removed the sentence “We previously reported that a low PNI was strongly associated with tumor progression and poor prognosis in patients with non-metastatic RCC” and added the sentence “Therefore, the PNI can aid clinicians in predicting the clinical outcomes of RCC and patients with low PNI need to be managed by nutritional support and treated in a way to correct malnutritional status.” In the discussion part of the manuscript. 

12) Conclusion. The conclusions are not supported by the results. They are not in line with the study aim that are to demonstrate the diagnostic accuracy of PNI as a prognostic factor for RCC

Answer

We thank the reviewer for this pertinent comment. To comply with the reviewer’s recommendations, we changed the sentence “In conclusion, the results of this study demonstrate that the PNI has value as a prognostic factor for RCC to “In conclusion, the results of this study demonstrate that diagnostic accuracy of the PNI as a prognostic factor for patients with RCC”.

13) References. They are up-to-date. However, several sentences in the manuscript lack references.

Answer

We thank the reviewer for this important comment. To comply with the reviewer’s recommendations, we added references in the sentences of the manuscript.

14) Table and Figures. They are of good quality.

Answer

We appreciate your trying to encourage us.

15) Please a native speacker check is straight reccomanded.

Answer

We thank the reviewer for this pertinent comment. However, our manuscript has already been qualified by an accredited institution. We attach the certificate file separately.

Reviewer #3:

1) Please rephrase the title and the text when you use the term "diagnostic" in association with "prognostic". For example in the title "diagnostic accuracy...as prognostic factor". Diagnosis and prognosis are different processes of medical evaluation. Please remove the term "diagnostic", in this study it has been evaluated the prognostic role of PNI.

Answer

We thank the reviewer for this important comment. To comply with the reviewer’s recommendations, we removed the words “diagnostic accuracy of” from the title. However, this study is diagnostic test accuracy meta analysis and it is different from general meta analsysis in a method for statistical analysis. Therefore, we think it would be better to leave the word in an essential part of the text.

2) A prognostic role has been proposed for other circulating biomarkers associated with different features of RCC biology, including carbonic anhydrase IX (CAIX), hypoxia-inducible factor-1α (HIF1α), CA15-3, PTX3, and C-reactive protein (CRP) (ref:PMID: 15126876; PMID: 24692843; PMID: 32345771;PMID: 20006861)

These studies should be referenced and discussed.

Answer

We thank the reviewer for this pertinent comment. To comply with the reviewer’s recommendations, we specifically discuss of circulating biomarkers associated with different features of RCC biology at the discussion section of the manuscript.

3) RCC is a metabolic disease characterized by a reprogramming of energetic metabolism. In particular the metabolic flux through glycolysis is partitioned (PMID: 30983433, PMID: 29371925, PMID: 28933387; PMID: 30538212), and mitochondrial bioenergetics and OxPhox are impaired , as well as lipid metabolism (PMID: 30538212; PMID: 32861643). In addition a recent study (PMID: 33322148) delineated a lipidomic profile of human ccRCC and integrated it with transcriptomic data to connect the variations in cancer lipid metabolism with gene expression changes. These findings should be referenced and discussed.

Answer

We thank the reviewer for this important comment. To comply with the reviewer’s recommendations, we more specifically discuss metabolism of RCC mentioned by reviewer in the discussion part of the manuscript.

---

## [Editor Report · Decision Letter 1]

8 Jul 2022

Prognostic Nutritional Index as a Prognostic Factor for Renal Cell Carcinoma: A Systematic Review and Meta-Analysis

PONE-D-22-03642R1

Dear Dr. Cho,

We’re pleased to inform you that your manuscript has been judged scientifically suitable for publication and will be formally accepted for publication once it meets all outstanding technical requirements.

Kind regards,

Giuseppe Lucarelli, M.D., Ph.D.

Academic Editor

PLOS ONE
---

## [Editor Report · Acceptance letter]

27 Jul 2022

PONE-D-22-03642R1 

Prognostic Nutritional Index as a Prognostic Factor for Renal Cell Carcinoma: A Systematic Review and Meta-Analysis 

Dear Dr. Cho:

I'm pleased to inform you that your manuscript has been deemed suitable for publication in PLOS ONE. Congratulations! Your manuscript is now with our production department. 

Kind regards, 

on behalf of

Dr. Giuseppe Lucarelli 

Academic Editor

PLOS ONE